# Adaptive Hierarchical Hyper-gradient Descent

## Abstract

In this study, we investigate learning rate adaption at different levels based on the hyper-gradient descent framework and propose a method that adaptively learns the optimizer parameters by combining different levels of adaptations. Meanwhile, we show the relationship between regularizing over-parameterized learning rates and building combinations of adaptive learning rates at different levels. The experiments on several network architectures, including feed-forward networks, LeNet-5 and ResNet-18/34, show that the proposed multi-level adaptive approach can significantly outperforms baseline adaptive methods in a variety of circumstances.

## 1 Introduction

The basic optimization algorithm for training deep neural networks is the gradient descent method (GD), which includes stochastic gradient descent (SGD), mini-batch gradient descent, and batch gradient descent. The model parameters are updated according to the first-order gradients of the empirical risks with respect to the parameters being optimized, while back-propagation is implemented for calculating the gradients of parameters (Ruder, 2016). Naïve gradient descent methods apply fixed learning rates without any adaptation mechanisms. However, considering the change of available information during the learning process, SGD with fixed learning rates can result in inefficiency and requires a large amount of computing resources in hyper-parameter searching. One solution is to introduce a learning rate adaptation. This idea can be traced back to the work on gain adaptation for connectionist learning methods (Sutton, 1992) and related extensions for non-linear cases (Schraudolph, 1999; Yu et al., 2006). In recent years, optimizers with adaptive updating rules were developed in the context of deep learning, while the learning rates are still fixed in training. The proposed methods include AdaGrad (Duchi et al., 2011), RMSProp (Tieleman and Hinton, 2012), and Adam (Kingma and Ba, 2015). In addition, there are optimizers aiming to address the convergence issue in Adam (Reddi et al., 2018; Luo et al., 2018) and to rectify the variance of the adaptive learning rate (Liu et al., 2019). Other techniques, such as *Lookahead*, can also achieve variance reduction and stability improvement with negligible extra computational cost (Zhang et al., 2019).

Even though the adaptive optimizers with fixed learning rates can converge faster than SGD in a wide range of tasks, the updating rules are designed manually while more hyper-parameters are introduced. Another idea is to use objective function information and update the learning rates as trainable parameters. These methods were introduced as automatic differentiation, where the hyper-parameters can be optimized with backpropagation (Maclaurin et al., 2015; Baydin et al., 2018). As gradient-based hyper-parameter optimization methods, they can be implemented as an online approach (Franceschi et al., 2017). With the idea of auto-differentiation, learning rates can be updated in real-time with the corresponding derivatives of the empirical risk (Almeida et al., 1998), which can be generated to all types of optimizers for deep neural networks (Baydin et al., 2017). Another step size adaptation approach called "L4", is based on the linearized expansion of the loss functions, which rescales the gradient to make fixed predicted progress on the loss (Rolinek and Martius, 2018). Furthermore, by addressing the issue of poor generalization performance of adaptive methods, dynamically bound for gradient methods was introduced to build a gradual transition between adaptive approach and SGD (Luo et al., 2018).

Another set of approaches train an RNN (recurrent neural network) agent to generate the optimal learning rates in the next step given the historical training information, known as "learning to learn"

(Andrychowicz et al., 2016). This approach empirically outperforms hand-designed optimizers in a variety of learning tasks, but another study has shown that it may not be effective for long horizons (Lv et al., 2017). The generalization ability of this approach can be improved by using meta training samples and hierarchical LSTMs (long short-term memory) (Wichrowska et al., 2017).

Beyond the adaptive learning rate, learning rate schedules can also improve the convergence of optimizers, including time-based decay, step decay, exponential decay (Li and Arora, 2019). The most fundamental and widely applied one is a piece-wise step-decay learning rate schedule, which could vastly improve the convergence of SGD and even adaptive optimizers(Luo et al., 2018; Liu et al., 2019). It can be further improved by introducing a statistical test to determine when to apply step-decay (Lang et al., 2019; Zhang et al., 2020). Also, there are works on warm-restart (O'donoghue and Candes, 2015; Loshchilov and Hutter, 2017), which could improve the performance of SGD anytime when training deep neural networks.

We find that the existing gradient or model-based learning rate adaptation methods including hyper-gradient descent, L4 and learning to learn only focus on global adaptation, which could be further extended to multi-level cases. That focus aims to introduce locally shared adaptive learning rates such as the layer-wise learning rate and parameter-wise learning rate and considers all levels' information in determining the updating step-size for each parameter. The main contribution of our study can be summarized as follows:

- We introduce hierarchical learning rate structures for neural networks and apply hyper-gradient descent to obtain adaptive learning rates at different levels.
- We introduce a set of regularization techniques for learning rates to address the balance of global and local adaptations and show the relationship with weighted combinations.
- We propose an algorithm implementing the combination of adaptive learning rates at multiple levels for model parameter updating.

## 2 MULTI-LEVEL ADAPTATION METHODS

### 2.1 LAYER-WISE, UNIT-WISE AND PARAMETER-WISE ADAPTATION

In the paper on hyper-descent (Baydin et al., 2017), the learning rate is set to be a scalar. However, to make the most of learning rate adaptation, in this study, we introduce layer-wise or even parameter-wise updating rules, where the learning rate $\boldsymbol{\alpha}_t$ in each iteration time step is considered to be a vector (layer-wise) or even a list of matrices (parameter-wise). For the sake of simplicity, we collect all the learning rates in a vector: $\boldsymbol{\alpha}_t = (\alpha_{1,t}, ..., \alpha_{N,t})^T$. Correspondingly, the objective $f(\boldsymbol{\theta})$ is a function of $\boldsymbol{\theta} = (\theta_1, \theta_2, ..., \theta_N)^T$, collecting all the model parameters. In this case, the derivative of the objective function $f$ with respect to each learning rate can be written as

$$\frac{\partial f(\boldsymbol{\theta}_{t-1})}{\partial \alpha_{i,t-1}} = \frac{\partial f(\theta_{1,t-1}, ..., \theta_{N,t-1})}{\partial \alpha_{i,t-1}} = \sum_{j=1}^{N} \frac{\partial f(\theta_{1,t-1}, ..., \theta_{N,t-1})}{\partial \theta_{j,t-1}} \frac{\partial \theta_{j,t-1}}{\partial \alpha_{i,t-1}}, \tag{1}$$

where $N$ is the total number of all the model parameters. Eq. (1) can be generalized to group-wise updating, where we associate a learning rate with a special group of parameters, and each parameter group is updated according to its only learning rate. Notice that although there is a dependency between $\alpha_{t-1}$ and $\theta_{t-2}$ with: $\alpha_{t-1} = \alpha_{t-2} - \beta \nabla f(\boldsymbol{\theta}_{t-2})$, where $\beta$ is the updating rate of hyper-gradient descent, we consider that $\alpha_{t-1}$ is calculated after $\theta_{t-2}$ and thus a change of $\alpha_{t-1}$ will not result in a change of $\theta_{t-2}$. Assume $\boldsymbol{\theta}_t = u(\boldsymbol{\Theta}_{t-1}, \alpha)$ is the updating rule, where $\boldsymbol{\Theta}_t = \{\boldsymbol{\theta}_s\}_{s=0}^{t}$ and $\alpha$ is the learning rate, then the basic gradient descent method for each group $i$ gives $\boldsymbol{\theta}_{i,t} = u(\boldsymbol{\Theta}_{t-1}, \alpha_{i,t-1}) = \boldsymbol{\theta}_{i,t-1} - \alpha_{i,t-1} \nabla_{\boldsymbol{\theta}_i} f(\boldsymbol{\theta}_{t-1})$. Hence for gradient descent

$$\frac{\partial f(\boldsymbol{\theta}_{t-1})}{\partial \alpha_{i,t-1}} = \nabla_{\boldsymbol{\theta}_i} f(\boldsymbol{\theta}_{t-1})^T \nabla_{\alpha_{i,t-1}} u(\boldsymbol{\Theta}_{t-1}, \alpha_{i,t-1}) = -\nabla_{\boldsymbol{\theta}_i} f(\boldsymbol{\theta}_{t-1})^T \nabla_{\boldsymbol{\theta}_i} f(\boldsymbol{\theta}_{t-2}). \tag{2}$$

Here $\alpha_{i,t-1}$ is a scalar with index $i$ at time step $t-1$, corresponding to the learning rate of the $i$th group, while the shape of $\nabla_{\boldsymbol{\theta}_i} f(\boldsymbol{\theta})$ is the same as the shape of $\boldsymbol{\theta}_i$. We particularly consider two special cases: (1) In **layer-wise adaptation**, $\boldsymbol{\theta}_i$ is the weight matrix of $i$th layer, and $\alpha_i$ is the particular learning rate for this layer. (2) In **parameter-wise adaptation**, $\boldsymbol{\theta}_i$ corresponds to a certain parameter involved in the model, which can be an element of the weight matrix in a certain layer.

## 2.2 Regularization on Adaptive Learning Rates

The selection of adaptation level should depend on a case-by-case basis. Global or parameter-wise adaptation is usually not the optimal choice across all circumstances. Recall that for deep neural networks, we typically use a relatively large architecture with regularization. This idea can also be applied to learning rate space with parameter structure. To address over-parameterization in implementing lower-level learning rate adaptation, we introduce regularization on learning rates to control the flexibility. First, for layer-wise adaptation, we can add the following regularization term to the loss function

$$L_{\text{lr\_reg\_layer}} = \lambda_{\text{layer}} \sum_l (\alpha_l - \alpha_g)^2, \tag{3}$$

where $l$ is the indices for each layer, $\lambda_{\text{layer}}$ is the layer-wise regularization coefficient, $\alpha_l$ and $\alpha_g$ are the layer-wise and global-wise adaptive learning rates. A larger $\lambda_{\text{layer}}$ can push each layer's learning rate towards the global learning rate across all the layers. Given a particular $\alpha_{g,t}$, the gradient of the loss function with respect to the learning rate $\alpha_l$ in layer $l$ can be written as

$$\frac{\partial L_{\text{full}}(\boldsymbol{\theta}, \alpha)}{\partial \alpha_{l,t}} = \frac{\partial L_{\text{model}}(\boldsymbol{\theta}, \alpha)}{\partial \alpha_{l,t}} + \frac{\partial L_{\text{lr\_reg}}(\boldsymbol{\theta}, \alpha)}{\partial \alpha_{l,t}}$$
$$= \nabla_{\boldsymbol{\theta}_l} f(\boldsymbol{\theta}_{t-1})^T \nabla_{\alpha_{l,t-1}} u(\boldsymbol{\Theta}_{t-2}, \alpha_{t-1}) + 2\lambda_{\text{layer}}(\alpha_{l,t} - \alpha_{g,t}). \tag{4}$$

Notice that the time step index of layer-wise regularization term is $t$ rather than $t-1$, which ensures that we push the layer-wise learning rates towards the corresponding global learning rates of the current step $t$. Denoting by $h_{l,t-1} = -\nabla_{\boldsymbol{\theta}_l} f(\boldsymbol{\theta}_{t-1})^T \nabla_{\theta_l} u(\boldsymbol{\Theta}_{t-2}, \alpha_{l,t-1})$, then the updating rule for learning rates can be written as

$$\alpha_{l,t} = \alpha_{l,t-1} - \beta \frac{\partial L_{\text{full}}(\boldsymbol{\theta}, \alpha)}{\partial \alpha_{l,t}} = \alpha_{l,t-1} - \beta(-h_{l,t-1} + 2\lambda_{\text{layer}}(\alpha_{l,t} - \alpha_{g,t})). \tag{5}$$

Eq. (5) has a close form solution but only applicable in the two-levels case. However, there is an extra hyper-parameter $\lambda_{\text{layer}}$ to be tuned. In addition, when there are more levels, components of learning rates at different levels can be interdependent. To construct a workable updating scheme for Eq. (5), we replace $\alpha_{l,t}$ and $\alpha_{g,t}$ with their relevant approximations. We take the strategy of using their updated version without considering regularization, i.e., $\hat{\alpha}_{l,t} = \alpha_{l,t-1} + \beta h_{l,t-1}$ and $\hat{\alpha}_{g,t} = \alpha_{g,t-1} + \beta h_{g,t-1}$, where $h_{g,t-1} = -\nabla_{\boldsymbol{\theta}} f(\boldsymbol{\theta}_{t-1})^T \nabla_{\alpha_{g,t-1}} u(\boldsymbol{\Theta}_{t-2}, \alpha_{g,t-1})$ is the global $h$ for all parameters. Here we regard $\hat{\alpha}_{l,t}$ and $\hat{\alpha}_{g,t}$ as the "virtual" layer-wise and global-wise learning rates for time step $t$ and taking them into the right-hand side of Eq. (5) gives the new updating rule as follows

$$\alpha_{l,t}^* = \alpha_{l,t-1} + \beta h_{l,t-1} - 2\beta\lambda_{\text{layer}}(\hat{\alpha}_{l,t} - \hat{\alpha}_{g,t}) = (1 - 2\beta\lambda_{\text{layer}})\hat{\alpha}_{l,t} + 2\beta\lambda_{\text{layer}}\hat{\alpha}_{g,t}. \tag{6}$$

Notice that in Eq. (6), the two terms are actually a weighted average of the layer-wise learning rate $\hat{\alpha}_{l,t}$ and global learning rate $\hat{\alpha}_{g,t}$ at the current time step. Since we hope to push the layer-wise learning rates towards the global one, the parameters should meet the constraint: $0 < 2\beta\lambda_{\text{layer}} < 1$, and thus they can be optimized using hyper-parameter searching within a bounded interval as well as gradient-based hyper-parameter optimizations. We can also consider the case where three levels of learning rate adaptations are involved, including global-wise, layer-wise, and parameter-wise adaptation. If we introduce two more regularization terms to control the variation of parameter-wise learning rate with respect to layer-wise learning rate and global learning rates, the regularization loss can be written as

$$L_{\text{lr\_reg\_para}} = \lambda_{\text{layer}} \sum_l (\alpha_l - \alpha_g)^2 + \lambda_{\text{para\_layer}} \sum_l \sum_p (\alpha_{pl} - \alpha_l)^2 + \lambda_{\text{para}} \sum_l \sum_p (\alpha_{pl} - \alpha_g)^2,$$

where $\alpha_{pl}$ is the learning rate for the $p$-th parameter inside layer $l$. The second and third terms push each parameter-wise learning rate towards the layer-wise learning rate and the global learning rates, respectively. Like the two-level case, the updating rule with this three-level regularization can be approximated by the weighted combination of three components under "virtual approximation". The detail of the updating rule for the three-levels case is given by Algorithm 1 in Section 2.3. We also provided a discussion on the bias of implementing "virtual approximation" in Appendix A.1.

In general, we can organize all the learning rates in a tree structure. For example, in the three-level case above, $\alpha_g$ will be the root node, while $\{\alpha_l\}$ are the children node at level 1 of the tree and $\{\alpha_{lp}\}$

are the children node of $\alpha_l$ as leaf nodes at level three of the tree. In a general case, we assume there are $L$ levels in the tree. Denote the set of all the paths from the root node to each of leave nodes as $\mathcal{P}$ and a path is denoted by $p = \{\alpha_1, \alpha_2, ..., \alpha_L\}$ where $\alpha_1$ is the root node, and $\alpha_L$ is the left node on the path. On this path, denote ancestors$(i)$ all the ancestor nodes of $\alpha_i$ along the path, i.e., ancestors$(i) = \{\alpha_1, ..., \alpha_{i-1}\}$. We will construct a regularizer to push $\alpha_i$ towards each of its parents. Then the regularization can be written as

$$L_{\text{lr\_reg}} = \sum_{p \in \mathcal{P}} \sum_{\alpha_i \in p} \sum_{\alpha_j \in \text{ancestor}(i)} \lambda_{ij}(\alpha_i - \alpha_j)^2. \tag{7}$$

Under this pair-wise $L_2$ regularization, the updating rule for any leave node learning rate $\alpha_L$ can be given by the following theorem whose proof is provided in Appendix A.2.

**Theorem 1.** *Under virtual approximation, the effect of applying pair-wise $L_2$ regularization Eq. (7) results in performing a weighted linear combination of virtual learning rates at different levels $\alpha_L^* = \sum_{j=1}^{L} \gamma_j \hat{\alpha}_j$ with $\sum_{j=1}^{L} \gamma_j = 1$, where each component $\hat{\alpha}_j$ is calculated by assuming no regularization.*

*Remarks:* Theorem 1 actually suggests that a similar updating rule can be obtained for the learning rate at any level on the path. All these have been demonstrated in Algorithm 1 for the three-level case.

## 2.3 PROSPECTIVE OF ADAPTIVE LEARNING RATE COMBINATION

Motivated by the analytical derivation in Section 2.2, we can consider combining adaptive learning rates at different levels as a substitute and approximation of regularization on the differences in learning rates. This makes the effect of learning rate regularization trainable with gradient-based methods. In a general form, assume that we have $L$ levels, which could include global-level, layer-level, unit-level and parameter-level, etc, Theorem 1 suggests the following updating rule: $\alpha_t = \sum_{j=1}^{L} \gamma_j \hat{\alpha}_{j,t}$. In a more general form, we can implement non-linear models such as neural networks to model the final adaptive learning rates with respect to the learning rates at different levels:$\alpha_t = g(\hat{\alpha}_{1,t}, \hat{\alpha}_{2,t}...\hat{\alpha}_{L,t}; \theta)$, where $\theta$ is the vector of parameters of the non-linear model. We can treat the combination weights $\{\gamma_1, ..., \gamma_L\}$ as trainable parameters, which can also be globally shared or parameter/layer-specific. In this study, we consider the globally shared combination weights. We only need these different levels of learning rate to have a hierarchical relationship to apply this method. For example, we can further introduce "filter level" to replace layer-level for the convolutional neural network if there is no clear layer structure, where the parameters in each filter will share the same learning rate.

As the real learning rates implemented in model parameter updating are weighted combinations, the corresponding gradient matrices cannot be directly used for learning rate updating. In this case, we first break down the gradient by the combined learning rate to three levels, use each of them to update the learning rate at each level, and then calculate the combination by the updated learning rates. Especially, $h_{p,t}$, $h_{l,t}$ and $h_{g,t}$ are calculated by the gradients of model losses without regularization, as is shown in Eq. (8)[1].

$$h_{p,t} = \frac{\partial f(\boldsymbol{\theta}, \alpha)}{\partial \alpha_{p,t}} = -\nabla_{\boldsymbol{\theta}} f(\boldsymbol{\theta}_{t-1}, \alpha)|_p \cdot \nabla_\alpha u(\boldsymbol{\Theta}_{t-2}, \alpha)|_p$$

$$h_{l,t} = \frac{\partial f(\boldsymbol{\theta}, \alpha)}{\partial \alpha_{l,t}} = -\text{tr}(\nabla_{\boldsymbol{\theta}} f(\boldsymbol{\theta}_{t-1}, \alpha)|_l^T \nabla_\alpha u(\boldsymbol{\Theta}_{t-2}, \alpha)|_l) \tag{8}$$

$$h_{g,t} = \frac{\partial f(\boldsymbol{\theta}, \alpha)}{\partial \alpha_t} = -\sum_{l=1}^{n} \text{tr}(\nabla_{\boldsymbol{\theta}} f(\boldsymbol{\theta}_{t-1}, \alpha)|_l^T \nabla_\alpha u(\boldsymbol{\Theta}_{t-2}, \alpha)_l)$$

where $h_t = \sum_l h_{l,t} = \sum_p h_{p,t}$ and $h_{l,t} = \sum_{p \in l\text{th layer}} h_p$ and $f(\theta, \alpha)$ corresponds to the model loss $L_{model}(\theta, \alpha)$ in Section 2.2. Algorithm 1 is the full updating rules for the newly proposed optimizer with three levels, which can be denoted as combined adaptive multi-level hyper-gradient descent (CAM-HD). In Algorithm 1, we introduce the general form of gradient descent based optimizers (Reddi et al., 2018; Luo et al., 2018): for SGD, $\phi_t(g_1, ...g_t) = g_t$ and $\psi_t(g_1, ...g_t) = 1$, while

---

[1]Here we use trace form to represent the sum of the element-wise product but compute in the simplest way.

---

**Algorithm 1:** Updating the rule of three-level CAM-HD

---

**input:** $\alpha_0, \beta, \delta, T$
**initialization:** $\theta_0, \gamma_{1,0}, \gamma_{2,0}, \gamma_{3,0}, \alpha_{p,0}, \alpha_{l,0}, \alpha_0, \alpha_{l,0}^* = \gamma_{1,0}\alpha_{p,0} + \gamma_{2,0}\alpha_{l,0} + \gamma_{3,0}\alpha_0$
**for** $t \in 1, 2, ..., T$ **do**

$\quad g_t = \nabla_\theta f(\theta, \alpha)$
$\quad$ **Update $h_{p,t}$, $h_{l,t}$ and $h_{g,t}$ by Eq. (8).**

$\quad \alpha_{p,t} = \alpha_{p,t-1} - \beta_p \frac{\partial f(\theta_{t-1})}{\partial \alpha_{p,t-1}^*} \frac{\partial \alpha_{p,t-1}^*}{\partial \alpha_{p,t-1}} = \alpha_{p,t-1} - \beta_p \gamma_{1,t-1} h_{p,t}$

$\quad \alpha_{l,t} = \alpha_{l,t-1} - \beta_l \sum_p \frac{\partial f(\theta_{t-1})}{\partial \alpha_{p,t-1}^*} \frac{\partial \alpha_{p,t-1}^*}{\partial \alpha_{l,t-1}} = \alpha_{l,t-1} - \beta_l \gamma_{2,t-1} \sum_p h_{p,t} = \alpha_{l,t-1} - \beta_l \gamma_{2,t-1} h_{l,t}$

$\quad \alpha_t = \alpha_{t-1} - \beta_g \sum_l \sum_p \frac{\partial f(\theta)}{\partial \alpha_{p,t-1}^*} \frac{\partial \alpha_{p,t-1}^*}{\partial \alpha_{t-1}} = \alpha_{t-1} - \beta_g \gamma_{3,t-1} h_{g,t}$

$\quad \alpha_{p,t}^* = \gamma_{1,t-1}\alpha_{p,t} + \gamma_{2,t-1}\alpha_{l,t} + \gamma_{3,t-1}\alpha_t$

$\quad \gamma_{1,t} = \gamma_{1,t-1} - \delta \frac{\partial L}{\partial \gamma_{1,t-1}} = \gamma_{1,t-1} - \delta \sum_p \frac{\partial L}{\partial \alpha_{p,t-1}^*} \frac{\partial \alpha_{p,t-1}^*}{\partial \gamma_{1,t-1}} = \gamma_{1,t-1} - \delta\alpha_{p,t-1} \sum_p \frac{\partial L}{\partial \alpha_{p,t-1}^*}$

$\quad \gamma_{2,t} = \gamma_{2,t-1} - \delta \frac{\partial L}{\partial \gamma_{2,t-1}} = \gamma_{2,t-1} - \delta \sum_p \frac{\partial L}{\partial \alpha_{p,t-1}^*} \frac{\partial \alpha_{p,t-1}^*}{\partial \gamma_{2,t-1}} = \gamma_{1,t-1} - \delta\alpha_{l,t-1} \sum_p \frac{\partial L}{\partial \alpha_{p,t-1}^*}$

$\quad \gamma_{3,t} = \gamma_{3,t-1} - \delta \frac{\partial L}{\partial \gamma_{3,t-1}} = \gamma_{3,t-1} - \delta \sum_p \frac{\partial L}{\partial \alpha_{p,t-1}^*} \frac{\partial \alpha_{p,t-1}^*}{\partial \gamma_{3,t-1}} = \gamma_{3,t-1} - \delta\alpha_{t-1} \sum_p \frac{\partial L}{\partial \alpha_{p,t-1}^*}$

$\quad \gamma_1 = \gamma_1/(\gamma_1 + \gamma_2 + \gamma_3), \gamma_2 = \gamma_1/(\gamma_1 + \gamma_2 + \gamma_3), \gamma_3 = \gamma_1/(\gamma_1 + \gamma_2 + \gamma_3)$
$\quad m_t = \phi_t(g_1, ...g_t)$
$\quad V_t = \psi_t(g_1, ...g_t)$
$\quad \theta_t = \theta_{t-1} - \alpha_{p,t}^* m_t/\sqrt{V_t}$

**end**
**return** $\theta_T$, $\gamma_{1,T}$, $\gamma_{2,T}$, $\gamma_{3,T}$, $\alpha_{p,T}$, $\alpha_{l,T}$, $\alpha_T$

---

for Adam, $\phi_t(g_1, ...g_t) = (1 - \beta_1)\Sigma_{i=1}^t \beta_1^{t-1} g_i$ and $\psi_t(g_1, ...g_t) = (1 - \beta_2)\text{diag}(\Sigma_{i=1}^t \beta_2^{t-1} g_i^2)$. Meanwhile, the corresponding $u(\Theta_{t-2}, \alpha)$ should be changed accordingly. Notice that in each updating time step of Algorithm 1, we re-normalize the combination weights $\gamma_1$, $\gamma_2$ and $\gamma_3$ to ensure that their summation is always 1 even after updating with stochastic gradient-based methods. An alternative way of doing this is to implement Softmax function. In addition, the training of $\gamma$s can also be extended to multi-level cases, which means we can have different combination weights in different layers. For the updating rates $\beta_p$, $\beta_l$ and $\beta_g$ of the learning rates at different levels, we set: $\beta_p = n_p\beta = \beta$, $\beta_l = n_l\beta$, $\beta_g = n\beta$, where $\beta$ is a shared parameter. This setting will make the updating steps of learning rates at different levels be on the same scale considering their difference in the number of parameters. An alternative way is to take the average based on the number of parameters in Eq. (8) at first.

## 2.4 CONVERGENCE ANALYSIS AND ALGORITHM COMPLEXITY

The proposed CAM-HD is not an independent optimization method, which can be applied in any gradient-based updating rules. Its convergence properties highly depend on the base optimizer that is applied. By referring to the discussion on convergence in (Baydin et al., 2017), we introduce $\kappa_{p,t} = \tau(t)\alpha_{p,t}^* + (1 - \tau(t))\alpha_\infty$, where the function $\tau(t)$ is selected to satisfy $t\tau(t) \to 0$ as $t \to \infty$, and $\alpha_\infty$ is a chosen constant value. Then we can demonstrate the convergence analysis for the three-level case in the following theorem.

**Theorem 2** (Convergence under mild assumptions about $f$)**.** *Suppose that $f$ is convex and $L$-Lipschitz smooth with $\|\nabla_p f(\theta)\| < M_p$, $\|\nabla_l f(\theta)\| < M_l$, $\|\nabla_g f(\theta)\| < M_g$ for some fixed $M_p$, $M_l$, $M_g$ and all $\theta$. Then $\theta_t \to \theta^*$ if $\alpha_\infty < 1/L$ where $L$ is the Lipschitz constant for all the gradients and $t \cdot \tau(t) \to 0$ as $t \to \infty$, where the $\theta_t$ are generated according to (non-stochastic) gradient descent.*

In the above theorem, $\nabla_p$ is the gradient of target function w.r.t. a model parameter with index $p$, $\nabla_l$ is the average gradient of target function w.r.t. parameters in a layer with index $l$, and $\nabla_g$ is the global average gradient of target function w.r.t. all model parameters. The proof of this theorem is given in Appendix A.3. Notice that when we introduce $\kappa_{p,t}$ instead of $\alpha_{p,t}^*$ in Algorithm 1, the corresponding gradients $\frac{\partial L(\theta)}{\partial \alpha_{p,t-1}^*}$ will also be replaced by $\frac{\partial L(\theta)}{\partial \kappa_{p,t-1}^*} \frac{\partial \kappa_{p,t-1}^*}{\partial \alpha_{p,t-1}^*} = \frac{\partial L(\theta)}{\partial \kappa_{p,t-1}^*} \tau(t)$.

We also made an analysis on the number of parameters and algorithm complexity (See Appendix A.4). Our method will not increase the number of model parameters but requires extra space complexity $\Delta T$ during training. Also it requires an extra time complexity but at least one-order smaller than training with baseline setting, while the absolute ratio is smaller than the inverse of batch size $\frac{\Delta T}{T} << 1/m_b$.

## 3 EXPERIMENTS

We use the feed-forward neural network models and different types of convolutions neural networks on multiple benchmark datasets to compare with existing baseline optimizers. For each learning task, the following optimizers will be applied: (a) standard baseline optimizers such as Adam and SGD; (b) hyper-gradient descent in (Baydin et al., 2017); (c) L4 stepsize adaptation for standard optimizers (Rolinek and Martius, 2018); (d) Adabound optimizer (Luo et al., 2018); (e) RAdam optimizer (Liu et al., 2019); and (f) the proposed adaptive combination of different levels of hyper-descent. The implementation of (b) is based on the code provided with the original paper. For each experiment, we provide the average curve and standard error bar in each time step with ten runs.

### 3.1 HYPER-PARAMETER TUNING

To compare the effect of CAM-HD with baseline optimizers, we first do hyperparameter tuning for the model training process with baseline optimizers by referring to related papers (Kingma and Ba, 2015; Baydin et al., 2017; Rolinek and Martius, 2018; Luo et al., 2018) as well as implementing an independent grid search. We mainly consider the hyper-parameters of batch size, learning rate, and momentum for models with different architecture. The search space for batch size is the set of $\{2^n\}_{n=3,...,9}$, while the search space for learning rate, hyper-gradient updating rate and combination weight updating rate (CAM-HD-lr) are $\{10^{-1}, ..., 10^{-4}\}$, $\{10^{-1}, ..., 10^{-10}\}$ and $\{10^{-1}, ..., 10^{-4}\}$ respectively. The selection criterion is the 5-fold cross-validation loss by early-stopping at the patience of 3. The optimized hyper-parameters for the tasks in this paper are given in Table 1. For the learning tasks with recommended learning rate schedules, we will apply these schedules as well.

Table 1: Hyperparameter Settings for Experiments

| Architecture | Dataset | Batch size | lr (SGD/SGDN) | lr (Adam) | Hyper-grad lr (SGD/SGDN) | Hyper-grad lr (Adam) | CAM-HD-lr |
|---|---|---|---|---|---|---|---|
| MLP 1 | | 32 | - | 0.0003 | - | 1.00E-07 | 0.01 |
| MLP 2 | MNIST | 64 | - | 0.001 | - | 1.00E-07 | 0.01 |
| MLP 3 | | 128 | - | 0.001 | - | 1.00E-07 | 0.01 |
| | MNIST | 256 | - | 0.001 | 1.00E-03 | 1.00E-08 | 0.03 |
| LeNet-5 | CIFAR10 | 256 | - | 0.001 | 1.00E-03 | 1.00E-08 | 0.03 |
| | SVHN | 128 | - | 0.001 | 1.00E-03 | 1.00E-08 | 0.03 |
| ResNet-18 | | 256 | 0.1 | 0.001 | 1.00E-06 | 1.00E-08 | 0.001 |
| ResNet-34 | CIFAR10 | 256 | 0.1 | 0.001 | 1.00E-06 | 1.00E-08 | 0.001 |

### 3.2 COMBINATION RATIO AND MODEL PERFORMANCES

First, we perform a study on the combination of different level learning rates. The simulations are based on image classification tasks on MNIST and CIFAR10 (LeCun et al., 1998; Krizhevsky and Hinton, 2012). One feed-forward neural network with three hidden layers of size [100, 100, 100] and two convolutional network models, including LeNet-5 (LeCun et al., 2015) and ResNet-18 (He et al., 2016), are implemented. We use full training sets of MNIST and CIFAR10 for training and full test sets for validation. In each case, two levels of learning rates are considered, which are the global and layer-wise adaptation for FFNN, and global and filter-wise adaptation for CNNs. Adam-CAM-HD optimizer is implemented in all three simulations. We change the combination weights of two levels in each case to see the change of model performance in terms of test classification accuracy at epoch 10, with the corresponding updating rate $\delta = 0$. Another hyper-parameter setting follows Table 1. We conduct ten runs at each combination weights with different parameter initializations for all three simulations and draw the error bars for standard errors. The result is given in Figure 1. We can see that in all three cases, the optimal performance is neither at full global level nor full layer/filter level, but a combination of two levels of adaptive learning rates. Still, the differences between the endpoints

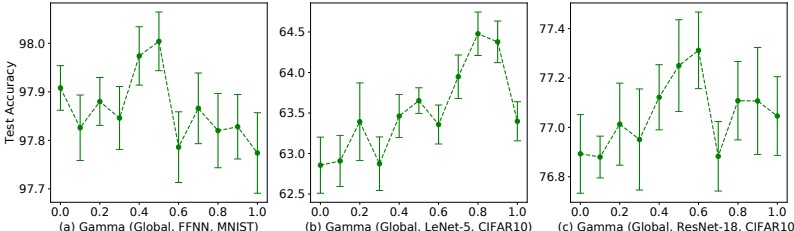

Figure 1: The diagram of model performances (at epoch 10) trained by Adam-CAM-HD with different fixed combination ratios in the case of two-level learning rates adaptation.

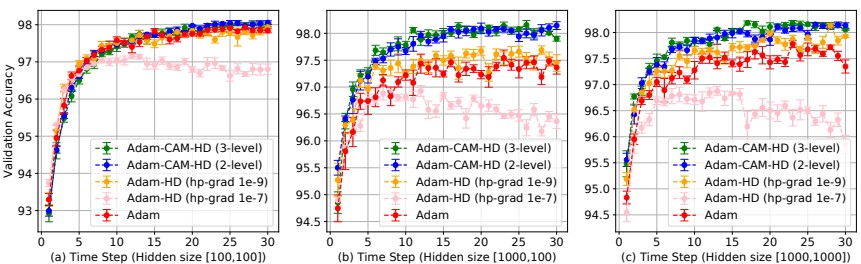

Figure 2: The comparison of learning curves of FFNN on MNIST with different adaptive optimizers.

and the optimal combination in terms of model performance have some statistical significance level. This supports our analysis in Section 2.2. Also, in real training processes, it is possible that the learning in favor of different combination weights in various stages and this requires the online updating of the combination weights.

### 3.3 FEED FORWARD NEURAL NETWORK FOR IMAGE CLASSIFICATION

This experiment is conducted with feed-forward neural networks for image classification on MNIST, including 60,000 training examples and 10,000 test examples. We use the full training set for training and the full test set for validation. Three FFNN with three different hidden layer configurations are implemented, including [100, 100], [1000, 100], and [1000, 1000]. Adaptive optimizers including Adam, Adam-HD with two hyper-gradient updating rates, and proposed Adam-CAM-HD are applied. For Adam-CAM-HD, we apply three-level parameter-layer-global adaptation with initialization of $\gamma_1 = \gamma_2 = 0.3$ and $\gamma_3 = 0.4$, and two-level layer-global adaptation with $\gamma_1 = \gamma_2 = 0.5$. Figure 2 shows the validation accuracy for different optimizers during the training process of 30 epochs. We can learn that both the two-level and three-level Adam-CAM-HD outperform the baseline Adam optimizer with optimized hyper-parameters significantly. For Adam-HD, we find that the default hyper-gradient updating rate $(10^{-7})$ for Adam applied in (Baydin et al., 2017) is not optimal in our experiments, while an optimized one of $10^{-9}$ can outperform Adam but still worse than Adam-CAM-HD with default hyper-gradient updating rate $(10^{-7})$.

### 3.4 LENET-5 FOR IMAGE CLASSIFICATION

The second experiment is done with LeNet-5, an early-year convolutional neural network without involving many building and training tricks. We compare a set of adaptive Adam optimizers including Adam, Adam-HD, Adam-CAM-HD, and L4 for the image classification learning task of MNIST, CIFAR10 and SVHN (Netzer et al., 2011). For Adam-CAM-HD, we apply a two-level setting with filter-wise and global learning rates adaptation and initialize $\gamma_1 = 0.2$, $\gamma_2 = 0.8$. We also implement an exponential decay function $\tau(t) = \exp(-rt)$ as was discussed in Section 2.4 with rate $r = 0.002$, while $t$ is the number of iterations. For L4, we implement the recommended L4 learning rate of 0.15. For Adabound and RAdam, we also apply the recommended hyper-parameters in the original

papers. The other hyper-parameter settings are optimized in Table 1. As we can see in Figure 3,

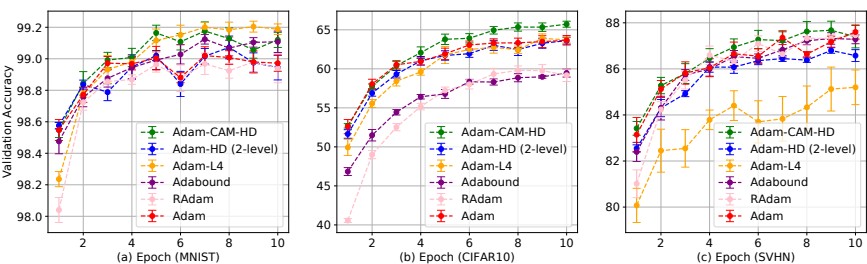

Figure 3: The comparison of learning curves of training LeNet-5 with different adaptive optimizers.

Adam-CAM-HD again shows the advantage over other methods in all the three sub-experiments, except MNIST L4 that could perform better in a later stage. The experiment on CIFAR10 and SVHN indicates that the recommended hyper-parameters for Adabound, RAdam and L4 could fail in some cases with unstable accuracy curves. On the other hand, Adam-HD can not significantly outperform Adam with the recommended and optimized hyper-gradient updating rate shared with Adam-CAM-HD. The corresponding summary of test performance is given in Table 2, in which the test accuracy of Adam-CAM-HD outperform other optimizers on both CIFAR10 and SVHN. Especially, it gives significantly better results than Adam and Adam-HD for all the three datasets.

Table 2: Summary of test performances with LeNet-5

|  | MNIST | | CIFAR10 | | SVHN | |
| --- | --- | --- | --- | --- | --- | --- |
|  | Test acc | Test S.E | Test acc | Test S.E | Test acc | Test S.E |
| Adam-CAM-HD | 98.93 | 0.07 | **65.55** | 0.18 | **87.58** | 0.37 |
| Adam-HD | 98.83 | 0.05 | 63.3 | 0.66 | 86.94 | 0.13 |
| Adam-L4 | **99.19** | 0.05 | 63.76 | 0.26 | 85.44 | 0.42 |
| Adabound | 99.11 | 0.05 | 59.79 | 0.70 | 87.22 | 0.14 |
| RAdam | 98.94 | 0.06 | 61.17 | 0.79 | 87.31 | 0.41 |
| Adam | 98.89 | 0.05 | 63.88 | 0.45 | 86.82 | 0.16 |

### 3.5 RESNET FOR IMAGE CLASSIFICATION

In the third experiment, we apply ResNets for image classification task on CIFAR10. We compare Adam and its adaptive optimizers, as well as SGD with Nestorov momentum (SGDN) and corresponding adaptive optimizers for training both ResNet-18 and ResNet-34. For SGDN methods, we apply a learning rate schedule, in which the learning rate is initialized to a default value of 0.1 and reduced to 0.01 or 10% (for SGDN-CAM-HD) after epoch 150. The momentum is set to be 0.9 for all SGDN methods. For Adam-CAM-HD SGDN-CAM-HD, we apply two-level CAM-HD with the same setting as the second experiment. In addition, we apply an exponential decay function with a decay rate $r = 0.001$. The validation accuracy results, training loss, and validation loss are shown in Figure 4. We can see that the validation accuracy of Adam-CAM-HD reaches about 90% in 40 epochs and consistently outperforms Adam, L4 and Adam-HD optimizers in a later stage. The L4 optimizer with recommended hyper-parameter and an optimized weight-decay rate of 0.0005 (instead of 1e-4 applied in other Adam-based optimizers) can outperform baseline Adam for both ResNet-18 and ResNet-34, while its training loss outperforms all other methods but with potential over-fitting. Adam-HD achieves better training loss than Adam after epoch 100. However, we find that the validation performance is not good with a default hyper-gradient coefficient of $10^{-8}$ (shared with Adam-CAM-HD). Instead, an optimized coefficient of $10^{-9}$ can make a safe but small improvement from Adam. RAdam performs slightly better than Adam-CAM-HD in terms of validation accuracy, but the validation cross-entropy of both RAdam and Adabound are worse than our method. Also, we find that in training ResNet-18/34, the validation accuracy and validation loss of SGDN-CAM-HD slightly outperform SGDN in most epochs even after the resetting of the learning rate at epoch 150.

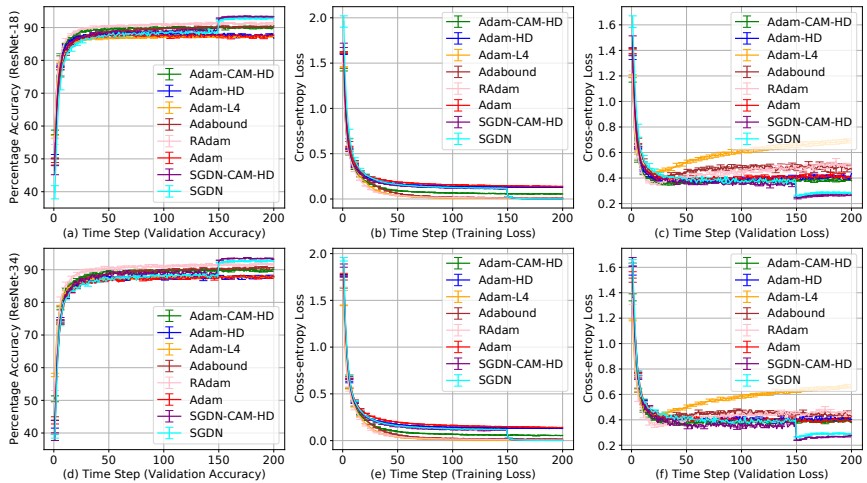

Figure 4: The learning curves of training ResNet on CIFAR10 with adaptive optimizers.

The test performances of different optimizers for ResNet-18 and ResNet-34 after 200 epoch of training are shown in Table 3. Notice that the results of SGDN and SGDN-CAM-HD are achieved with a piece-wise constant learning rates schedule, while the results of Adam-based optimizers are achieved without a learning rate schedule. We can learn that the proposed CAM-HD method can improve the corresponding baseline method (Adam, Adam-HD and SGDN)

Table 3: Summary of test performances with ResNet-18/34

|  | ResNet-18 | | ResNet-34 | |
| --- | --- | --- | --- | --- |
|  | Test acc | Test S.E | Test acc | Test S.E |
| Adam | 86.94 | 0.13 | 87.87 | 0.2 |
| Adam-HD | 87.26 | 0.35 | 88.48 | 0.48 |
| Adam-L4 | 87.81 | 0.22 | 88.02 | 0.15 |
| Adabound | 90.29 | 0.15 | 90.15 | 0.30 |
| RAdam | 91.54 | 0.17 | 91.76 | 0.28 |
| Adam-CAM-HD | 90.10 | 0.23 | 90.18 | 0.06 |
| SGDN | 93.04 | 0.23 | 92.93 | 0.19 |
| SGDN-CAM-HD | **93.2** | 0.24 | **93.47** | 0.23 |

with statistical significance in almost every case. Adam-CAM-HD performs a bit worse than RAdam but comparable to Adabound in terms of the average test accuracy for both ResNet-18 and ResNet-34. As a higher-level adaptation method, CAM-HD can be applied on top of RAdam/Adabound/L4 for further improvement.

## 4 CONCLUSION

In this study, we propose a gradient-based learning rate adaptation strategy by introducing hierarchical multiple-level learning rates in deep neural networks. By considering the relationship between regularization and the combination of adaptive learning rate at different levels, we further propose a joint algorithm for adaptively learning each level's combination weight. Experiments on FFNN, LeNet-5, and ResNet-18/34 indicate that the proposed methods can outperform the standard ADAM/SGDN and other baseline methods with statistical significance. Although the advantage is not fully guaranteed, our method achieves a higher adaptation level and can be continuously reduced to baseline methods under a specific set of hyper-parameters. This could bring more thoughts and further study on implementing a hierarchical learning rate system for deep neural networks.

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

# A   APPENDIX FOR PAPER: ADAPTIVE MULTI-LEVEL HYPER-GRADIENT DESCENT

## A.1   ANALYSIS ABOUT VIRTUAL APPROXIMATION

Consider the difference between Eq. (5) and Eq. (6) in the paper:

$$\alpha_{l,t}^{*} - \alpha_{l,t} = -2\beta\lambda_{\text{layer}}((\hat{\alpha}_{l,t} - \hat{\alpha}_{g,t}) - (\alpha_{l,t} - \alpha_{g,t})). \tag{9}$$

Based on the setting of multi-level adaptation, on the right-hand side of Eq. (9), global learning rate is updated without regularization $\hat{\alpha}_{g,t} = \alpha_{g,t}$. For the layer-wise learning rates, the difference is given by $\hat{\alpha}_{l,t} - \alpha_{l,t} = 2\beta\lambda_{\text{layer}}(\alpha_{l,t} - \alpha_{g,t})$, which corresponds to the gradient with respect to the regularization term. Thus, Eq. (9) can be rewritten as:

$$\alpha_{l,t}^{*} - \alpha_{l,t} = -2\beta\lambda_{\text{layer}}(2\beta\lambda_{\text{layer}}(\alpha_{l,t} - \alpha_{g,t})) = -4\beta^2\lambda_l^2(1 - \frac{\alpha_{g,t}}{\alpha_{l,t}})\alpha_{l,t} \tag{10}$$

which is the error of the virtual approximation introduced in Eq. (6) in the paper. If $4\beta^2\lambda_l^2 << 1$ or $\frac{\alpha_{g,t}}{\alpha_{l,t}} \to 1$, this approximation becomes accurate. Another way for handling Eq. (4) in the paper is to implement the previous-step learning rates in the regularization term.

$$\alpha_{l,t} \approx \alpha_{l,t-1} - \beta(-h_{l,t-1} + 2\lambda_{\text{layer}}(\alpha_{l,t-1} - \alpha_{g,t-1})). \tag{11}$$

Since we have $\alpha_{l,t} = \hat{\alpha}_{l,t} - 2\beta\lambda_{\text{layer}}(\alpha_{l,t} - \alpha_{g,t})$ and $\hat{\alpha}_{l,t} = \alpha_{l,t-1} + \beta h_{l,t-1}$, using the learning rates in the last step for regularization will introduce a higher variation from term $\beta h_{l,t-1}$, with respect to the true learning rates in the current step. Thus, we consider the proposed virtual approximation works better than last-step approximation.

## A.2   PROOF OF THEOREM 1:

*Proof.* Consider the learning regularizer

$$L_{\text{lr\_reg}}(\alpha) = \sum_{p\in\mathcal{P}}\sum_{\alpha_i\in p}\sum_{\alpha_j\in\text{parents}(i)}\lambda_{ij}(\alpha_i - \alpha_j)^2. \tag{12}$$

To apply hyper-gradient descent method to update the learning rate $\alpha_L$ at level $L$, we need to work out the derivative of $L_{\text{lr\_reg}}$ with respect to $\alpha_L$. The terms in Eq. (12) involving $\alpha_L$ are only $(\alpha_i - \alpha_j)^2$ where $\alpha_j$ is an ancestor on the path from the root to the leave node $\alpha_L$. Hence

$$\frac{\partial L_{\text{full}}(\boldsymbol{\theta},\alpha)}{\partial \alpha_{L,t}} = \frac{\partial L_{\text{model}}(\boldsymbol{\theta},\alpha)}{\partial \alpha_{L,t}} + \frac{\partial L_{\text{lr\_reg}}(\alpha)}{\partial \alpha_{L,t}}$$

$$= -\nabla_{\boldsymbol{\theta}_L}f(\boldsymbol{\theta}_{t-1})^T\nabla_{\boldsymbol{\theta}_L}u(\Theta_{t-2},\alpha_{t-1}) + \sum_{\alpha_j\in\text{acenstors}(L)}2\lambda_{Lj}(\alpha_{L,t} - \alpha_{j,t}). \tag{13}$$

As there are exactly $L-1$ ancestors on the path, we can simply use the index $j = 1,2,...,L-1$. The corresponding updating function for $\alpha_{n,t}$ is:

$$\alpha_{L,t} = \alpha_{n,t-1} - \beta(h_L + \sum_{j=1}^{L-1}2\lambda_{Lj}(\alpha_{L,t} - \alpha_{j,t}))$$

$$\approx \hat{\alpha}_{L,t}(1 - 2\beta\sum_{j=1}^{L-1}\lambda_{Lj}\alpha_{n,t}) + \sum_{j=1}^{L-1}(2\beta\lambda_{Lj}\hat{\alpha}_{j,t})) \tag{14}$$

$$= \sum_{j=1}^{L}\gamma_j\hat{\alpha}_{j,t}.$$

where

$$\gamma_L = 1 - 2\beta\sum_{j=1}^{L-1}\lambda_{Lj}, \tag{15}$$

$$\gamma_j = 2\beta\lambda_{Lj}, \quad \text{for } j = 1,2,...,L-1.$$

This form satisfies $\alpha_L^* = \sum_{j=1}^{L}\gamma_j\hat{\alpha}_j$ with $\sum_{j=1}^{L}\gamma_j = 1$. This completes the proof. □

## A.3 Proof of Theorem 2:

*Proof.* We take three-level's case discussed in Section 2 for example, which includes global level, layer-level and parameter-level. Suppose that the target function $f$ is convex, L-Lipschitz smooth at all levels, which means for all $\theta_1$ and $\theta_2$:

$$
\begin{aligned}
||\nabla_p f(\theta_1) - \nabla_p f(\theta_2)|| &\leq L_p ||\theta_1 - \theta_2|| \\
||\nabla_l f(\theta_1) - \nabla_l f(\theta_2)|| &\leq L_l ||\theta_1 - \theta_2|| \\
||\nabla_g f(\theta_1) - \nabla_g f(\theta_2)|| &\leq L_g ||\theta_1 - \theta_2|| \\
L &= \max\{L_p, L_l, L_g\}
\end{aligned}
\tag{16}
$$

and its gradient with respect to parameter-wise, layer-wise, global-wise parameter groups satisfy $||\nabla_p f(\theta)|| < M_p$, $||\nabla_l f(\theta)|| < M_l$, $||\nabla_g f(\theta)|| < M_g$ for some fixed $M_p$, $M_l$, $M_g$ and all $\theta$. Then the effective combined learning rate for each parameter satisfies:

$$
\begin{aligned}
|\alpha_{p,t}^*| &= |\gamma_{p,t-1}\alpha_{p,t} + \gamma_{l,t-1}\alpha_{l,t} + \gamma_{g,t-1}\alpha_t| \\
&\leq (\gamma_{p,t-1} + \gamma_{l,t-1} + \gamma_{g,t-1})\alpha_0 + \beta \sum_{i=0}^{t-1} \left( \gamma_{p,t-1} n_p \max_p \{|\nabla f(\theta_{p,i+1})^T \nabla f(\theta_{p,i})|\} \right. \\
&\quad + \gamma_{l,t-1} n_l \max_l \{|\nabla f(\theta_{l,i+1})^T \nabla f(\theta_{l,i})|\} + \gamma_{g,t-1}|\nabla f(\theta_{g,i+1})^T \nabla f(\theta_{g,i})| \bigg) \\
&\leq \alpha_0 + \beta \sum_{i=0}^{t-1} \left( \gamma_{p,t-1} n_p \max_p \{||\nabla f(\theta_{p,i+1})|| ||\nabla f(\theta_{p,i})||\} \right. \\
&\quad + \gamma_{l,t-1} n_l \max_l \{||\nabla f(\theta_{l,i+1})|| ||\nabla f(\theta_{l,i})||\} + \gamma_{g,t-1}||\nabla f(\theta_{g,i+1})|| ||\nabla f(\theta_{g,i})|| \bigg) \\
&\leq \alpha_0 + t\beta(n_p M_p^2 + n_l M_l^2 + M_g^2)
\end{aligned}
\tag{17}
$$

where $\theta_{p,i}$ refers to the value of parameter indexed by $p$ at time step $i$, $\theta_{l,i}$ refers to the set/vector of parameters in layer with index $l$ at time step $i$, and $\theta_{g,i}$ refers to the whole set of model parameters at time step $i$. In addition, $n_p$ and $n_l$ are the total number of parameters and number of the layers, and we have applied $0 < \gamma_p, \gamma_l, \gamma_g < 1$. This gives an upper bound for the learning rate in each particular time step, which is $O(t)$ as $t \to \infty$. By introducing $\kappa_{p,t} = \tau(t)\alpha_{p,t}^* + (1 - \tau(t))\alpha_\infty$, where the function $\tau(t)$ is selected to satisfy $t\tau(t) \to 0$ as $t \to \infty$, so we have $\kappa_{p,t} \to \alpha_\infty$ as $t \to \infty$. If $\alpha_\infty < \frac{1}{L}$, for larger enough $t$, we have $1/(L+1) < \kappa_{p,t} < 1/L$, and the algorithm converges when the corresponding gradient-based optimizer converges for such a learning rate under our assumptions about $f$. This follows the discussion in (Karimi et al., 2016; Sun, 2019). $\square$

## A.4 Algorithm Complexity

### A.4.1 Number of parameters and space complexity

The proposed adaptive optimizer is for efficiently updating the model parameters, while the final model parameters will not be increase by introducing CAM-HD optimizer. However, during the training process, several extra intermediate variables are introduced. For example, in the discussed three-level's case for feed-forward neural network with $n_{\text{layer}}$ layers, we need to restore $h_{p,t}$, $h_{l,t}$ and $h_{g,t}$, which have the sizes of $S(h_{p,t}) = \sum_{l=1}^{n_{\text{layer}}-1}(n_l + 1)n_{l+1}$, $S(h_{l,t}) = n_{\text{layer}}$ and $S(h_{g,t}) = 1$, respectively, where $n_i$ is the number of units in $i$th layer. Also, learning rates $\alpha_{p,t}$, $\alpha_{l,t}$, $\alpha_{g,t}$ and take the sizes of $S(a_{p,t}) = \sum_{l=1}^{n_{\text{layer}}-1}(n_l + 1)n_{l+1}$, $S(a_{l,t}) = n_{\text{layer}}$, $S(a_{g,t}) = 1$, $S(a_{g,t}) = 1$, and $S(a_{p,t}^*) = \sum_{l=1}^{n_{\text{layer}}-1}(n_l + 1)n_{l+1}$, respectively. Also we need a small set of scalar parameters to restore $\gamma_1$, $\gamma_2$ and $\gamma_3$ and other coefficients.

Consider the fact that in training the baseline models, we need to restore model parameters, corresponding gradients, as well as the intermediate gradients during the implementation of chain rule, CAM-HD will take twice of the space for storing intermediate variables in the worst case. For two-level learning rate adaptation considering global and layer-wise learning rates, the extra space complexity by CAM-HD will be one to two orders' smaller than that of baseline model during training.

A.4.2 TIME COMPLEXITY

In CAM-HD, we need to calculate gradient of loss with respect to the learning rates at each level, which are $h_{p,t}$, $h_{l,t}$ and $h_{g,t}$ in three-level's case. However, the gradient of each parameter is already known during normal model training, the extra computational cost comes from taking summations and updating the lowest-level learning rates. In general, this cost is in linear relation with the number of differentiable parameters in the original models. Here we discuss the case of feed-forward networks and convolutional networks.

Recall that for feed-forward neural network the whole computational complexity is:

$$T(n) = O(m \cdot n_{\text{iter}} \cdot \sum_{l=2}^{n_{\text{layer}}} n_l \cdot n_{l-1} \cdot n_{l-2}) \tag{18}$$

where $m$ is the number of training examples, $n_{\text{iter}}$ is the iterations of training, and $n_l$ is the number of units in the $l$-th layer. On the other hand, when using three-level CAM-HD with, where the lowest level is parameter-wise, we need $n_{\text{layer}}$ element products to calculate $h_{p,t}$ for all layers, one $n_{\text{layer}}$ matrix element summations to calculate $h_{l,t}$ for all layers, as well as a list summation to calculate $h_{g,t}$. In addition, two element-wise summations will also be implemented for calculating $\alpha_{p,t}$ and $\alpha_p^*$. Therefore, the extra computational cost of using CAM-HD is $\Delta T(n) = O(n_b \cdot n_{\text{iter}} \sum_{l=2}^{n_{\text{layer}}} (n_l \cdot n_{l-1} + n_l))$, where $n_b$ is the number of mini-batches for training. Notice that $m_b = m/n_b$ is the batch size, which is usually larger than 100. This extra cost is more than one-order smaller than the computational complexity of training a model without learning rate adaptation. For the cases when the lowest level is layer-wise, only one element-wise matrix product is needed in each layer to calculate $h_{l,t}$. For convolutional neural networks, we have learned that the total time complexity of all convolutional layers is (He and Sun, 2015):

$$O(n_b \cdot n_{\text{iter}} \cdot \sum_{l=1}^{n_{conv\_layer}} (n_{l-1} \cdot s_l^2 \cdot n_l \cdot m_l^2)) \tag{19}$$

where $l$ is the index of a convolutional layer, and $n_{conv\_layer}$ is the depth (number of convolutional layers). $n_l$ is the number of filters in the $l$-th layer, while $n_{l-1}$ is known as the number of input channels of the $l$-th layer. $s_l$ is the spatial size of the filter. $m_l$ is the spatial size of the output feature map. If we consider convolutional filters as layers, the extra computational cost for CAM-HD in this case is $\Delta T(n) = O(n_b \cdot n_{\text{iter}} \sum_{l=1}^{n_{conv\_layer}} ((n_{l-1} \cdot s_l^2 + 1) \cdot n_l))$, which is still more than one order smaller than the cost of model without learning rate adaptation.

Therefore, for large networks, applying CAM-HD will not significantly increase the computational cost from the theoretical prospective.

## A.5 SUPPLEMENTARY EXPERIMENTAL RESULTS

### A.5.1 LEARNING OF COMBINATION WEIGHTS

The following figures including Figure 5, Figure 6, Figure 7 and Figure 8 give the learning curves of combination weights with respect to the number of training iterations in each experiments, in which each curve is averaged by 5 trials with error bars. Through these figures, we can compare the updating curves with different models, different datasets and different CAM-HD optimizers.

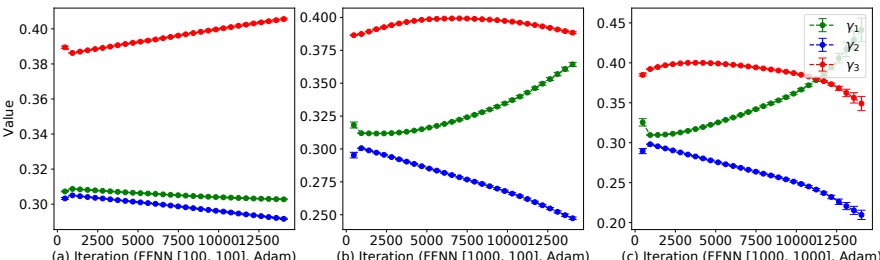

Figure 5: Learning curves of $\gamma$s for FFNN on MNIST with Adam.

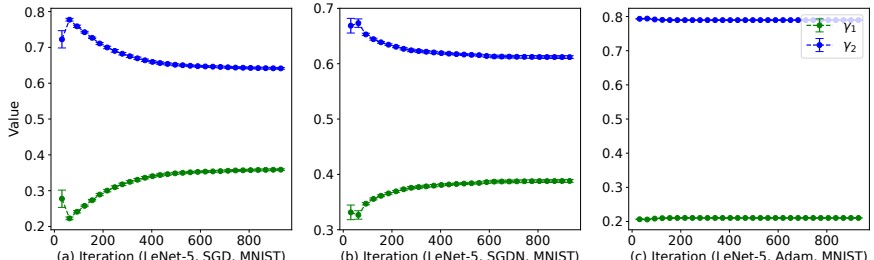

Figure 6: Learning curves of $\gamma$s for LeNet-5 on MNIST with SGD, SGDN and Adam.

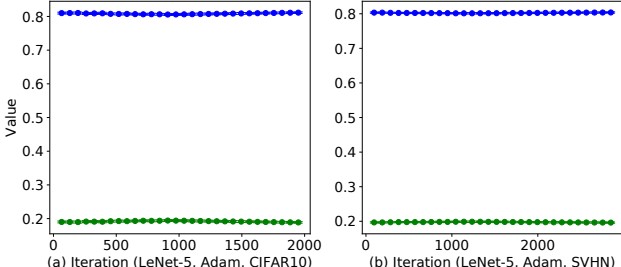

Figure 7: Learning curves of $\gamma$s for LeNet-5 on CIFAR10 and SVHN.

Figure 5 corresponds to the experiment of FFNN on MNIST in Section 3.3, which is a three-level case. We can see that for different FFNN architecture, the learning behaviors of $\gamma$s also show different patterns, although trained on a same dataset. Meanwhile, the standard errors for multiple trials are much smaller relative to the changes of the average combination weight values.

Figure 6 corresponds to the learning curves of $\gamma$s in the experiments of LeNet-5 for MNIST image classification with SGD, SGDN and Adam, which are trained on 10% of original training dataset.

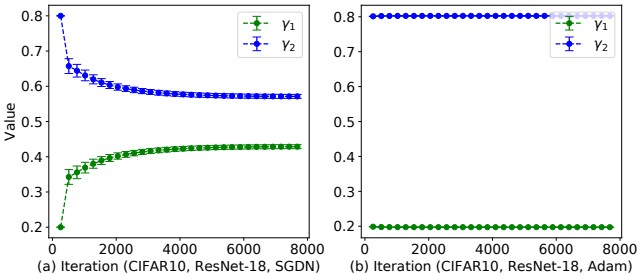

Figure 8: Learning curves of $\gamma$s for ResNet-18.

In addition, Figure 7 corresponds to the learning curves of $\gamma$s in the experiments of LeNet-5 for CIFAR10 and SVHN image classification with Adam-CAM-HD.

As is shown in Figure 6, for SGD-CAM-HD, SGDN-CAM-HD and Adam-CAM-HD, the equilibrium values of combination weights are different from each other. Although the initialization $\gamma_1 = 0.2$, $\gamma_2 = 0.8$ and the updating rate $\delta = 0.03$ are set to be the same for the three optimizers, the values of $\gamma_1$ and $\gamma_2$ only change in a small proportion when training with Adam-CAM-HD, while the change is much more significant towards larger filter/layer-wise adaptation when SGD-CAM-HD or SGDN-CAM-HD is implemented. The numerical results show that for SGDN-CAM-HD, the average value of weight for layer-wise adaptation $\gamma_1$ jumps from 0.2 to 0.336 in the first epoch, then drop back to 0.324 before keeping increasing till about 0.388. For Adam-CAM-HD, the average $\gamma_1$ moves from 0.20 to 0.211 with about 5% change. In Figure 7, both the two subplots are models trained with Adam-CAM-HD. For the updating curves in Figure 7(a), which is trained on CIFAR10 with Adam-CAM-HD, the combination weight for filter-wise adaptation moves from 0.20 to 0.188. Meanwhile, for the updating curves in Figure 7(b), which is trained on SVHN, the combination weight for filter-wise adaptation moves from 0.20 to 0.195.

The similar effect can also be observed from the learning curves of $\gamma$s for ResNet-18, which is given in Figure 8 and we only take the first 8,000 iterations. Again, we find that in training ResNet-18 on CIFAR10, the combination weights of SGD/SGDN-CAM-HD change much faster than that of Adam-CAM-HD. There are several reasons for this effect: First, in the cases when $\gamma$s do not move significantly, we apply Adam-CAM-HD, where the main learning rate (1e-3) is only about 1%-6% of the learning rate of SGD or SGDN (1e-1). In Algorithm 1, we can see that the updating rate of $\gamma$s is in proportion of alpha given other terms unchanged. Thus, for the same tasks, if the same value of updating rate $\delta$ is applied, the updating scale of $\gamma$s for Adam-CAM-HD can be much smaller than that for SGDN-CAM-HD. Second, this does not mean that if we apply a much larger $\delta$ for Adam-CAM-HD, the combination weights will still not change significantly or the performance will not be improved. It simply means that using a small $\delta$ can also achieve good performance due to the goodness of initialisation points. Third, it is possible that Adam requires lower level of combination ratio adaptation for the same network architecture compared with SGD/SGDN due to the fact that Adam itself involves stronger adaptiveness.

### A.5.2 OTHER EXPERIMENTAL RESULTS

In Figure 2, Figure 3 and Figure 4 of the paper, we have shown the curves of validation accuracies to compare different adaptive optimizers in a variety of learning tasks. Here we further provide the training and validation cross-entropy loss curves for corresponding methods in these tasks. Figure 8 is the full results of FFNNs, and Figure 9 is the results of LeNet-5.

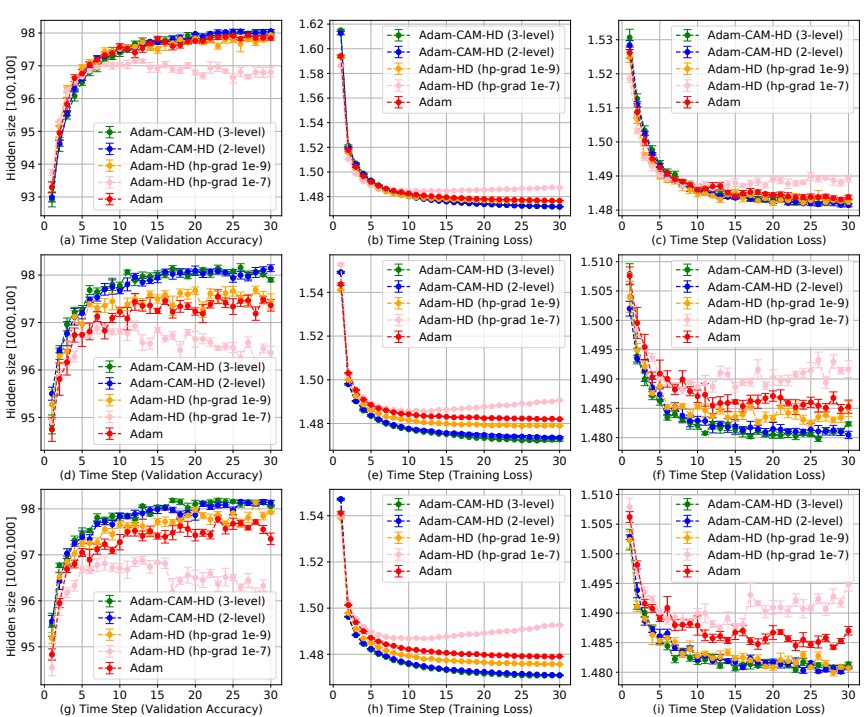

Figure 9: The comparison of learning curves of FFNN on MNIST with different adaptive optimizers.

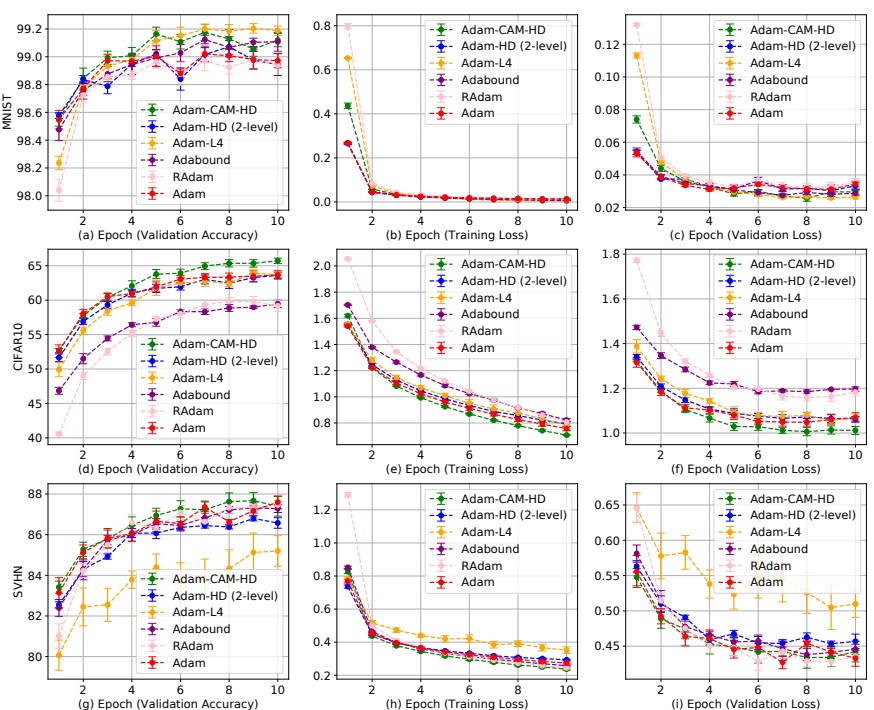

Figure 10: The comparison of learning curves of training LeNet-5 on MNIST, CIFAR10 and SVHN with different adaptive optimizers.

