# OpenReview forum: "Adaptive Hierarchical Hyper-gradient Descent"
_ICLR.cc/2021/Conference — Reject_

### Official Review · AnonReviewer2 · 2020-10-28
**Extending hyper-gradient descent to many levels of learning rates**

**Rating:** 5
**Confidence:** 2

**Review:**

### Summary of the paper

The paper investigates the setting of hyper-gradient descent in the context of adapting learning rates at different levels in a neural network (e.g. per layer). The paper derives an equivalence between regularization of such learning rates and a weighted combination of non-regularized adapted learning rates. Experiments on small datasets and small network architectures are performed and show an improvement over baselines including Adam and SGD.

### Main review

After the introduction (which is well-written) I found the paper relatively hard to follow. Sections 2.2 and 2.3 are long and introduce a large variety of options that *could* be used without giving the reader a clear perspective of where this discussion is going. In the experiments, a lot of different combinations of small networks and datasets are used, and while the learning curves look promising, it is hard to assess how robustly the improvements may transfer to other settings, because e.g. the parameters $\gamma$ are initialized differently in the experiments. The experiment of Fig.1 shows that the variation in the measurements can be very high with respect to $\gamma$, such that the effect of the algorithm as such is a bit hard to assess. (And e.g. Fig.7 seems to indicate that the $\gamma$s sometimes do not move from their initial values.) This makes me wonder how much of an improvement it is to go from optimizing hyper-parameters of e.g. SGDN or Adam to optimizing "hyper-hyper-parameters".

When reading the paper I also wondered if any relation between the behavior of the adapted learning rates and other quantities in training (e.g. momentum per parameter) can be established. Does the adaptation counteract or increase momentum effects? Or does this depend on other quantities? Any such connection could provide insight and/or help with intuitions about the adaptation behavior. I did not see any attempt into such a direction in the paper.

The relation between regularization and weighting of unregularized values (Theorem 1) seems interesting.

As the experiments go as far as ResNet-34 on CIFAR10, it would be interesting if a comparison to existing results from the literature could be made. E.g. the accuracies in Fig.4 seem to go into the area of 92-93%, which seems appropriate for this combination, but it is hard to tell from the graph alone. (It would be even more impressive if a clear statement could be made that shows that the final accuracy is better than for the optimizers used in a work that is (close to) state of the art (under given constraint like e.g. architecture=ResNet-18).

A (brief) discussion of the additional computational cost (in terms of memory and operation count) in the main body text would be useful.

* Pros: The paper takes hyper-gradient to many new multi-levels.
* Cons: The paper is somewhat hard to follow, and it is not immediately clear how transferable the experimental results are to other settings.

### Minor details and comments
* The font in the figures is very small in some cases, e.g. Figure 5 is very hard to read even when zooming in considerably on the screen.
* Typos/nits:
  * p.3 "leave nodes" -> leaf
  * p.4 "use each of them to updated the learning rate" -> update
  * p.6 "tunning"
  * p.8 "in later stage"; "our methods achieves" -> method
  * in several places: CMA -> CAM
  * Lowercasing in reference titles, e.g. "Rmsprop", "smd gain"

---

> ### Author Response · Authors · 2020-11-19
> **Response to AnonReviewer2**
>
> Thanks a lot for the reviewer's detailed feedback. We would like to answer your concerns as follows.
>
> *C1: The paper relatively hard to follow after the introduction.*
>
> A1: Thanks for letting us know. We will try to make it more clear in the updated version.
>
> *C2: It is hard to assess how robustly the improvements may transfer to other settings, because e.g. the parameters γ are initialized differently in the experiments. The experiment of Fig.1 shows that the variation in the measurements can be very high with respect to γ, such that the effect of the algorithm as such is a bit hard to assess. (And e.g. Fig.7 seems to indicate that the γs sometimes do not move from their initial values.)*
>
> A2: We tune the hyper-parameter for our method for each experiment mainly because we need to adjust them in the correct scale. Within a range, small changes of the hyper-parameters do not harm the performance and fixed hyper-parameter settings are transferable for similar tasks with same baseline optimizer. For example, in the experiment of feed-forward networks, we initialise the  combination weights to (0.3,0.3,0.4). If we use (0.333,0.333,0.334) instead, we can still achieve an improvement of performance, as the combination weights will be updated during training.
>
> In the experiment in Figure 1, we do not update the γs, which means we use fixed combination weights to see if there is a significant difference on the test accuracy at epoch 10. If the optimal combinations are not at endpoints, it means that a combination of layer/filter-wise adaptation and global adaptation with certain weights can outperform both full layer-wise and full global adaptation. In this case, it is necessary to apply a combination of different levels of adaptive learning rates rather than implementing a pure single-level adaptation.
>
> When we apply same updating rates for γs, sometime it moves in large scale, but sometime it does not change that much (The numerical value still changes by 1 or 2 percent. For example, in Figure 6 (c), the numerical value of γ1 moves from 0.20 to 0.2104 on average, and  γ2 moves from 0.80 to 0.7896; In Figure 7 (a), γ1 moves from 0.20 to 0.1886). There are several reasons: First, in the cases when γs do not move significantly, we apply Adam optimizer, where the main learning rate alpha is only about 1% of the learning rate of SGD. In Algorithm 1, we can see that the updating rate of γs is in proportion of alpha given other terms unchanged. Thus, for same tasks, if same value of delta is applied, the updating rate of γs for Adam can be much smaller than that for SGD. Second, this does not mean that if we apply a much larger delta, the combination weights will still not change significantly or the performance will not be improved. It simply means that using a small delta can also achieve good performance due to the goodness of initialisation points.
>
> *C3: Does the adaptation counteract or increase momentum effects? Or does this depend on other quantities?*
>
> A3: Thanks for your suggestions. Our method is developed from the basic hyper-gradient descent (Baydin et al. 2017) for learning rate adaptation. It can be applied to any kind of gradient-based methods such as SGD and Adam. For different optimizers, the updating rules for learning rates varies accordingly. We can consider the case of parameter-wise learning rate adaptation for SGD by referring to Eq.(2) in the paper. The comparison can be shown as follows:
>
> SGD:
> $w_t = w_{t-1}-a *dw_{t-1}$
>
> SGD-momentum:
> $[w_t = w_{t-1}-a*v_t = w_{t-1}-a*(b*v_{t-1}+(1-b)*dw_{t-1})$
>
> SGD-HD:
> $w=w-a'*dw = w-(a+b*dw_{t-1}*dw_{t-2})*dw_{t-1}$ (the simplest case)
>
> SGD-momentum-HD:
> $w=w-a'*dw = w-(a-b*dw_{t-1}*du_{t-1}/da)*v_t = w-(a+b*dw_{t-1}*(b*v_{t-1}+(1-b)*dw_{t-2}))*(b*v_{t-1}+(1-b)*dw_{t-1})$
>
> where w can be considered as a particular parameter. If the gradients of the previous two steps $dw_{t-1}, dw_{t-2}$ are large and with the same sign, the learning rate for that parameter will also increase in updating. This brings some kind of momentum accumulating effect. However, the end-to-end gradient updating rule for learning rates adaptation can take use of more information from the basement optimizers than traditional momentum method. If momentum term exists in the basement optimizers (e.g. SGDN and Adam), the corresponding parameter-wise adaptive rule will be derived accordingly, with both momentum and learning rate adaptation involved.
>
> There are also auto differentiation methods for learning rate adaptation taking account of second-order information or all historical gradients (Maclaurin et al. 2015), where the proposed CAM-HD can also be applied if the adaptations in different levels are obtainable.

---

> > ### Author Response · Authors · 2020-11-20
> > **Response to AnonReviewer2 [2/2]**
> >
> > *C4: As the experiments go as far as ResNet-34 on CIFAR10, it would be interesting if a comparison to existing results from the literature could be made.*
> >
> > A4: Thanks for your suggestion. We think there are a lot of version of open-source code for architecture like ResNet-18. We compare our results with that in L4 paper (Rolinek et al. NeurIPS 2018), where ResNet-34 with Adam gives an accuracy of about 86% and manually tuned learning rate schedule achieves about 92%. Also, in (Liu et al. 2019), their results on CIFAR10 with ResNet-20 are around 91-92%. Some achieves higher accuracies but probably with more data pre-processing, advanced learning rate schedules, extra tricks and tuning, or training with a much longer period of time. In fact, optimal hyper-parameter setting also depends on other tricks, number of training epochs, or even random seeds. In our study, we focus on standard ResNets with 200 epochs of training. As we are comparing our method with a set of other adaptive optimizers such as L4 and Adam-HD, it is quite hard to guarantee that each of them can be tuned to its optimal setting when advanced tricks (not even applied in the original paper of the corresponding method) is introduced.
> >
> > *Cons*
> >
> > *C5: The paper is somewhat hard to follow, and it is not immediately clear how transferable the experimental results are to other settings.*
> >
> > A5: The hyper-parameter setting for each experiment is transferable to similar tasks, although the optimal setting can shift a bit on a case-by-case basis. It is safe to apply a relatively small updating rates to achieve a relatively small improvement.
> >
> > *C6: Minor details and comments.*
> >
> > A6: Thanks for pointing out these. We will revise it in the updated version.
> >
> > Reference:
> >
> > [1] Baydin, Atilim Gunes, et al. "Online learning rate adaptation with hypergradient descent." arXiv preprint arXiv:1703.04782 (2017).
> >
> > [2] Rolinek, M., & Martius, G. (2018). L4: Practical loss-based stepsize adaptation for deep learning. In Advances in Neural Information Processing Systems (pp. 6433-6443).
> >
> > [3] Liu, L., Jiang, H., He, P., Chen, W., Liu, X., Gao, J., & Han, J. (2019, September). On the Variance of the Adaptive Learning Rate and Beyond. In International Conference on Learning Representations.
> >
> > [4] Maclaurin, D., Duvenaud, D., & Adams, R. (2015, June). Gradient-based hyperparameter optimization through reversible learning. In International Conference on Machine Learning (pp. 2113-2122).

---

> > > ### Comment · AnonReviewer2 · 2020-11-24
> > > **Thank you for your response**
> > >
> > > Thank you for your response and for the updates to the paper. I have read your response and will take it into account in the following discussions.

---

> > > > ### Author Response · Authors · 2020-11-25
> > > > **Thanks for evaluating our response**
> > > >
> > > > Thank you!

---

### Official Review · AnonReviewer4 · 2020-10-29
**This paper proposed a hierarchical learning rate setting method for network training.**

**Rating:** 5
**Confidence:** 4

**Review:**

Setting appropriate learning rate for network optimization is an important task in deep learning applications. This paper investigates the setting of learning rates for network parameters in different levels, e.g., individual parameter, each layer and global levels. By setting the constraints on the learning rates at multiple scales, the paper derived a hierarchical learning rate setting approach, which is the combination of adaptive learning rates at different levels.

Overall, this proposed learning rate setting method seems to be interesting, however, I have some concerns on the setting of the hyper-parameters of the proposed method.

1. The paper proposed to set the constraints at different levels by Eqn. (7). I have concern on how to automatically set the hyper-parameters of combination weights.  The details on setting / learning these hyper-parameters should be clarified in a more clear way.

2. What are the possible reasons that the proposed hierarchical learning rate can improve the baseline optimizer Adam and SGD?

3. The proposed learning rates setting method can be combined with any gradient-based network optimizer. More combinations and corresponding results and comparisons should be given to show how much the proposed technique can improve the different baseline gradient-based optimization methods.

4. The comparisons with more network optimizers should be given in the experiments.


----
Post rebuttal comments：
Thanks for the responses from the authors. These responses partially solved my questions. I think that the initialization of hyper-parameters of combination weights seem to be heuristic, and it is unclear on the effects/robustness of its initialization on the optimization performance. My questions on more comparisons and more combinations with other optimizers are not well answered.

I also read the other reviews and responses, I agree with other reviewers on the concerns of experiments, justifications, robustness, etc. Considering that it needs some additional works to solve all these concerns, I suggest the authors to improve the paper and submit it to one following conference.

---

> ### Author Response · Authors · 2020-11-19
> **Response to AnonReviewer4**
>
> Thanks a lot for your feedback and suggestions. For your concerns, we would like to make responses as follows:
>
> *C1: The paper proposed to set the constraints at different levels by Eqn. (7). I have concern on how to automatically set the hyper-parameters of combination weights. The details on setting / learning these hyper-parameters should be clarified in a more clear way.*
>
> A1: We selected the initialized weights based on some exploratory experiments such as those given in Figure 1, where we change the fixed combination ratios (without updating) of two levels to see the change of model performance. It is not difficult to find a good initialisation by 2 or 3 trials. If there is no prior knowledge, we can directly initialize them equally to 1/n, where n is the number of components/levels. The algorithm can learn the combination ratios by updating them during the training process, which is provided in Algorithm 1 (updating of gammas).
>
> *C2: What are the possible reasons that the proposed hierarchical learning rate can improve the baseline optimizer Adam and SGD?*
>
> A2: One possible reason is that it introduces a technique that improves the level of adaptiveness but controls the level of over-parameterization by reducing the variation of adaptive learning rates. Usually a higher level of adaptiveness corresponds to a higher level of parameterization. However, in our method the adaptiveness is introduced on regularization terms of different levels of adaptive learning rate,  which controls rather than increases the variations of adaptive learning rates (especially when parameter-wise adaptive learning rates are applied).
>
> *C3: The proposed learning rates setting method can be combined with any gradient-based network optimizer. More combinations and corresponding results and comparisons should be given to show how much the proposed technique can improve the different baseline gradient-based optimization methods.*
>
> A3: Thanks for your suggestion. This paper just applies the method with Adam and SGDN to show its effectiveness. In our opinion, an improvement for one baseline optimizer can be useful enough given the optimizer is widely applied. We notice that in the original paper of Adabound (Luo et al. 2018), they only implemented Adabound and AMSBound in their experiment.  In the paper of L4 (Rolinek et al. 2018), they only implemented L4-Adam and L4-momentum in the experiment. Thus, we will only consider to combine our method with more baseline gradient-based optimization methods in the experiments if there is sufficient time.
>
> *C4: The comparisons with more network optimizers should be given in the experiments.*
>
> A4: In fact, as the proposed method is a higher-level adaptation approach, it can be applied for any type of gradient-based network optimizer. The original contents of our paper only compare it with baseline methods such as Adam and SGD, as well as gradient-based adaptation methods such as Hyper-gradient descent and L4. Usually we think Adam is better than momentum and RMSProp in terms of convergence speed, SGD with learning rate schedule is better than other traditional optimizers to achieve good generalisation performance, and some other optimizers only works better than Adam/SGD under particular circumstances (e.g. AMSGrad, Adabound, etc). The tuning for advanced optimizers requires more time and computing power (some of them outperformed by baseline Adam with original hyper-parameter setting), especially for architectures applied in our paper but not applied in their papers.  We will add the experiments with more network optimizers for comparisons in the updated version if time permitted.
>
> Reference:
>
> [1] Luo, L., Xiong, Y., Liu, Y., & Sun, X (2018). Adaptive Gradient Methods with Dynamic Bound of Learning Rate. In International Conference on Learning Representations.
>
> [2] Rolinek, M., & Martius, G. (2018). L4: Practical loss-based stepsize adaptation for deep learning. In Advances in Neural Information Processing Systems (pp. 6433-6443).
>
> [3] Liu, L., Jiang, H., He, P., Chen, W., Liu, X., Gao, J., & Han, J. (2019, September). On the Variance of the Adaptive Learning Rate and Beyond. In International Conference on Learning Representations.

---

### Official Review · AnonReviewer1 · 2020-10-31
**Limited novelty and weak empirical justification**

**Rating:** 5
**Confidence:** 3

**Review:**

Update: I really appreciate the response from the authors. Some of my original concerns have been addressed, and additional experiments help to show the benefits of CAM-HD, so I have increased my score to 5. But, after reading other reviews and responses, I still believe that this work needs to be compared to advanced learning rate adaptation methods. Most reviewers have pointed out the presentation and insufficient experiments, so it's better to submit the improved version to one of upcoming conferences.

**Summary**
This work proposes an optimizer that adaptively determines a learning rates from different levels (global, layer-wise, parameter-wise) based on the hypergradient framework.  The proposed optimizer introduces many additional hyperparameters, and empirical evidence is not strong compared to baselines.

**Detailed comments**

The proposed method adaptively adjusts learning rates at different levels (parameter-wise, layer-wise, and global). This needs to be compared to previous learning rate schedulers, but I found that only a basic scheduler has been compared. How about the performance of CAM-HD compared to previous approaches?
* Using Statistics to Automate Stochastic Optimization, NeurIPS’19.
* Statistical Adaptive Stochastic Gradient Methods, Arxiv’20. (This work is an extension of the neurips paper above).
* Large Batch Training of Convolutional Networks (Arxiv’17). This work also proposes a ayer-wise Adaptive Rate Scaling.
* SGDR: Stochastic Gradient Descent with Warm Restarts, ICLR’17.

This seems to be very sensitive to newly introduced hyperparameters (beta and gammas). This method requires extensive grid search to find a suitable beta, because the optimal beta is different across tasks (as in Table 1).

The proposed optimizer has been validated only on small-scale datasets. It’s difficult to predict how to behave in training on a large-scale dataset (such as ImageNet). In addition, I’m not quite sure that this approach works well with strong data augmentation techniques (for instance, auto augmentation). Also, I guess that the proposed scheduler will not be working well on large-batch settings (for instance, batch size >= 1K).

“For the learning tasks with recommended learning rate schedules, we will apply these schedules as well.” -> In experiments, just a simple step decay lr rule has been applied. What about using cosine annealing lr schedulers? It could improve the baseline consistently.

---

> ### Author Response · Authors · 2020-11-19
> **Response to AnonReviewer1  [1/2]**
>
> Thanks a lot for the reviewer's feedback and we would like to answer the concerns as follows.
>
> *C1: The proposed method adaptively adjusts learning rates at different levels (parameter-wise, layer-wise, and global). This needs to be compared to previous learning rate schedulers.*
>
> A1: Thanks for recommending the related papers to us. As far as we know, manually designed learning rate schedule is a different type of method from automated learning rate adaptation. Published related papers on learning rate adaptation usually only compare their method with baseline method (Andrychowicz et al. 2016, Baydin et al. 2017), a simple learning rate schedule (Rolinek et al. 2018) or apply a learning rate schedule along with adaptive optimizers (Luo et al. 2019). In fact, CAM-HD can also apply learning rate schedules in many ways to achieve further improvement (e.g. apply piece-wise adaptive scheme, replace global level lr with scheduled lr while adapt combination weights and other levels, etc). One example is our ResNet experiment on CIFAR10 with SGDN and SGDN-CAM-HD.
>
> We think it is interesting to compare or combine our methods with advanced learning rate schedules in follow-up work, but it is also important to focus the comparing the same type of methods given the limited time and resources. We would like to clarify that in our method, layer-wise adaptive learning rate is only one component of the weight combination.
>
> *C2: This seems to be very sensitive to newly introduced hyperparameters (beta and gammas). This method requires extensive grid search to find a suitable beta, because the optimal beta is different across tasks (as in Table 1).*
>
> A2: The optimal newly introduced hyper-parameter may shift but is not very sensitive. Table 1 shows the optimised values of betas and updating rate for combination weights, but a wider range of values also work well. The hyper-parameters are transferable to other tasks (especially similar tasks) with improvements from baselines, although the improvements may not be optimal or at the highest level without tuning. To our knowledge, most of the optimizers require extra tuning of hyper-parameters when first introduced. The original paper of hyper-gradient descent (Baydin et al. 2017) also tuned beta for each type of model. For the proposed CAM-HD method, it is safe to apply 1e-8 as beta for Adam-CAM-HD to make an improvement in most of the cases. As for gammas, they can be updated during the training process with Algorithm 1.
>
> *C3: The proposed optimizer has been validated only on small-scale datasets. It’s difficult to predict how to behave in training on a large-scale dataset (such as ImageNet). In addition, I’m not quite sure that this approach works well with strong data augmentation techniques (for instance, auto augmentation). Also, I guess that the proposed scheduler will not be working well on large-batch settings (for instance, batch size >= 1K).*
>
> A3: We are not sure if our methods can significantly outperform baseline optimisers on very large datasets with the same hyper-parameter settings. One issue is that we do not have that amount of computing power to run the experiment. However, the improvement on relatively small datasets also make sense, because we do not always work on large datasets and lots of applications for deep learning models are on small-datasets. Also, we usually consider batch-size as a hyper-parameter to be tuned, so the method is not necessary to work very well on large batch settings if small batch size can result in good performance. We believe every method or optimizer has the condition for performing well, and the selection of method can be done accordingly. In addition, our method can reduce to baseline method by setting small updating rates and large combination weight for global level. Thus, a relatively small updating rate can safely make an improvement (although not optimal) in most of cases.
>
> *C4: In experiments, just a simple step decay lr rule has been applied. What about using cosine annealing lr schedulers? It could improve the baseline consistently.*
>
> A4: Advanced learning rate schedules can be effective in further improving the performance of optimizers. However, we find that usually learning rate schedules can improve both the baseline methods and the proposed method. It is unfair to only apply advanced learning rate schedule to baselines but not ours. We will discuss more learning rate schedules in the introduction part of the updated version and may add corresponding experiments if time permitted. Current we think it is only necessary to apply other learning rate schedules if the study is to propose a new learning rate schedule rather than a new gradient-based adaptive method. We notice that in a lot of related paper such as Adabound (Luo et al. 2019), L4 (Rolinek et al. 2018) and RAdam (Liu et al. 2019), also only the step decay lr is applied.

---

> > ### Author Response · Authors · 2020-11-20
> > **Our Response to AnonReviewer1 [2/2]**
> >
> > Reference:
> >
> > [1] Baydin, A. G., Cornish, R., Rubio, D. M., Schmidt, M., & Wood, F. (2018, February). Online Learning Rate Adaptation with Hypergradient Descent. In International Conference on Learning Representations.
> >
> > [2] Andrychowicz, M., Denil, M., Gomez, S., Hoffman, M. W., Pfau, D., Schaul, T., ... & De Freitas, N. (2016). Learning to learn by gradient descent by gradient descent. In Advances in neural information processing systems (pp. 3981-3989).
> >
> > [3] Luo, L., Xiong, Y., Liu, Y., & Sun, X (2018). Adaptive Gradient Methods with Dynamic Bound of Learning Rate. In International Conference on Learning Representations.
> >
> > [4] Rolinek, M., & Martius, G. (2018). L4: Practical loss-based stepsize adaptation for deep learning. In Advances in Neural Information Processing Systems (pp. 6433-6443).
> >
> > [5] Liu, L., Jiang, H., He, P., Chen, W., Liu, X., Gao, J., & Han, J. (2019, September). On the Variance of the Adaptive Learning Rate and Beyond. In International Conference on Learning Representations.

---

### Official Review · AnonReviewer3 · 2020-11-09
**Ambiguous analysis and results from a novel method**

**Rating:** 5
**Confidence:** 3

**Review:**

Quality

+ The authors have published code for replicability.
+ The baseline evaluations are of reasonable quality.
+ The proofs and theorems around convergence and complexity are high quality.
- The results on their metrics are poor and ambiguous. They only evaluate on validation accuracy / cross entropy, which do not improve substantially.
- Combination ratio results look inconsistent across datasets & models.
- The analysis is limited and there are few generalizable insights to be gleaned from the paper.
- There’s no transfer analysis or generality analysis, implying that each task will have to have its hyper-parameters tuned independently.
- It’s not demonstrated that overparameterization is a problem. Networks are already overparameterized without issue.

Clarity
Their combination ratios plot is unclear. What should be taken away from these changes in Gamma? The analysis is unclear. Convergence analysis is unclear. There are plenty of typos.


Originality

The regularization methodology is somewhat novel, to my knowledge. Hypergradients are known and this is an extension to more parameters which will interact with one another. Multiple levels of tree based interacting hyperparameters is novel, to my knowledge.

Significance

This result isn’t a substantial improvement over existing methods. There aren’t clear insights to be gleaned that will generalize to other work on hypergradients (other than that it’s possible to try to regularize with a hierarchical parameter scheme).
This may be evidence that trying to tune every parameter simultaneously is overkill / too challenging.



Pros

The authors present a novel method for learning hyperparameters of a model at multiple levels which typically are not manually tuned. The potential upside of a working method here is high, as the learning rate dramatically impacts model performance. They publish code for reproducibility. The authors propose and implement a novel regularization method for their new hyperparameters as well. There are proofs backing convergence claims made by the authors. The baselines have been tuned and the presentation is honest.

Cons

The paper’s results are ambiguous. The paper isn’t carefully written - it’s unclear what ‘levels of adaptations’ refer to until after the introduction. It’s not clear what conclusions should be drawn from the combination ratios graph as gamma changes (Figure 1). The paper’s analysis value is limited, it’s unclear what solid & general insights about hypergradients can be taken away from it.



Other feedback:

Typo in the abstract: combination -> combinations
Typo in the first line of introduction: is gradient -> is the gradient
Typo in the second sentence of the second paragraph of the introduction: function -> functions
Typo in the third paragraph of the introduction: horizon -> horizons
Typo in first paragraph of Experiments: lowercase p in ‘The Proposed -> The proposed’
Likely better not to shorten ‘Feed Forward Neural Network’ to FFNN in the section heading (though this is fine in the body of the text).

---

> ### Author Response · Authors · 2020-11-19
> **Response to AnonReviewer3 [Part 1/2]**
>
> Thanks very much for the reviewer's feedback. We would like to answer (A) the reviewer's concerns (C) as follows.
>
> *Quality:*
>
> *C1:  The results on their metrics are poor and ambiguous. They only evaluate on validation accuracy / cross entropy, which do not improve substantially.*
>
> A1: We do not agree with the reviewer's comment that our results are poor and ambiguous. Based on our experiments on feed-forward neural networks, LeNet-5 and ResNet 18/34, the proposed methods do provide improvements in many cases with statistical significance. As far as we know, there is little chance for new optimizers to substantially outperform the state-of-the-art optimisers by a large margin on validation/test sets. An improvement with statistical significance will be sufficiently meaningful. We have also provided the results of test accuracies and corresponding standard errors for different optimizers in the updated version.
>
> *C2: Combination ratio results look inconsistent across datasets & models.*
>
> A2: In fact, the optimal combination ratio can be trained by updating the combination weights following Algorithm 1. Figure 1 shows the test performance after 10 epochs of training when we apply fixed combination weights without updating. It indicates a weighted combination of multiple levels of adaptive learning rates can outperform the case when we only apply a single level of learning rates adaptation such as global adaptation or layer-wise adaptation. Thus, effective learning of the combination weights is important.
>
> *C3: The analysis is limited and there are few generalizable insights to be gleaned from the paper.*
>
> A3: We will try to improve the analysis in the updated version. Traditional learning rate adaptation methods consider parameter-wise adaptation such as Adam, or global adaptations (in a higher level) such as hyper-gradient descent or L4. The combination of multiple levels adaptive learning rate with hierarchical structure can be considered as an original innovation in our study, which can be applied on top of any gradient-based optimizers. The experiments do indicate improvements that are statistical significant can be achieved although not by a large margin.
>
> *C4: There’s no transfer analysis or generality analysis, implying that each task will have to have its hyper-parameters tuned independently.*
>
> A4: In our experiment, we find that the performance improvement does not require tuning the hyper-parameters independently if the task or model is similar. For example, the hyper-gradient updating rate for LeNet-5, ResNet-18 and ResNet-34 are all set to be 1e-8 in our experiments no matter the dataset being learned.  Also, the hyper-parameter CAM-HD-lr is shared among the each group of models (FFNNs, LeNet-5, ResNets) for all datasets being learned. In fact, we notice that the selected hyper-parameters are transferable to similar tasks for an improvement, while the optimal hyper-parameter setting can shift a bit.
>
> As far as we know, most of the optimisers for deep neural networks require tuning the related hyper-parameters (e.g. learning rate) for different tasks. Although methods like L4 and Adam-HD can achieve some level of learning rate adaptation, we find that they do not always perform well without tuning the introduced parameters or other hyper-parameters such as weight decay rate. If a small effort of tuning can provide a statistical significant improvement of model performance, we will consider the method to be useful.
>
> *C5: It’s not demonstrated that over-parameterization is a problem. Networks are already over-parameterized without issue.*
>
> A5: Our experiments show that the optimal combination weights of different levels in usually not at the endpoints, which means that simply applying parameter-wise hyper-gradient descent or global hyper-gradient descent is not the optimal choice. For networks, we usually introduce regularization on the model parameters to avoid over-fitting, while for adaptive learning rates, we also need to control the level of over-parameterization.

---

> > ### Author Response · Authors · 2020-11-19
> > **Our Response to AnonReviewer3 [Part 2/2]**
> >
> > *Clarity*
> >
> > *C6: Their combination ratios plot is unclear. What should be taken away from these changes in Gamma? The analysis is unclear. Convergence analysis is unclear. There are plenty of typos.*
> >
> > A6: Thanks for pointing out this. We apologize for the typos that affect reading and will make revision on that. In fact, as we put the change of Gammas in appendix, we just want to show the learning behaviours of them for different models and learning tasks. More discussion on this will be added in the updated version.
> >
> > *Significance*
> >
> > *C7: This result isn’t a substantial improvement over existing methods. There aren’t clear insights to be gleaned that will generalize to other work on hyper-gradients (other than that it’s possible to try to regularize with a hierarchical parameter scheme).*
> >
> > A7: The improvements are with statistical significance in a variety of circumstances. The hierarchical learning rate scheme achieves higher-level adaptiveness compared with single-level case such as global adaptation or parameter-wise adaptation, while it reduces over-parameterization with respect to lower-level adaptation such as parameter-wise adaptation. Notice that usually a higher-level adaptiveness requires a higher-level parameterization, but the extra adaptiveness introduced is for regularization effect that controls the variation of parameter-wise adaptive learning rates.
> >
> > *Cons*
> >
> > *C8: The paper’s results are ambiguous. The paper isn’t carefully written - it’s unclear what ‘levels of adaptations’ refer to until after the introduction. It’s not clear what conclusions should be drawn from the combination ratios graph as gamma changes (Figure 1). The paper’s analysis value is limited, it’s unclear what solid & general insights about hyper-gradients can be taken away from it.*
> >
> > A8: Thanks for pointing out these issues. We will improve our written. In fact, Figure 1 demonstrates that models trained by a fixed weighted combination of multiple levels of adaptive learning rate (with optimized combination ratio) can outperform the case when a single-level adaptation approach is applied (e.g global learning rate adaption or layer-wise learning rate adaption). To find the optimal combination weights during the training process, the corresponding gradient-based learning algorithm is introduced.

---

### Author Response · Authors · 2020-11-24
**Main updates in the final version**

Thanks again to all the reviewers for their time and efforts in reviewing our paper and providing valuable feedbacks. We have uploaded our revised version with the following changes:

1. Following the suggestion of AnonReviewer1, we provided a literature review on related works of learning rate schedules in the introduction part.

2. Following the suggestion of AnonReviewer4, we added the experiments of more adaptive optimizers including Adabound and RAdam for LeNet-5, ResNet-18 and ResNet-34.

3. Following the comments of AnonReviewer3, we provided the summary tables for test accuracies with standard errors for each experiment.

4. Following the comments of AnonReviewer2 and AnonReviewer3, we made more discussion on the learning curves of combination weights given in Figure 5 to Figure 8.

5. Following the suggestion of AnonReviewer2, we added a paragraph in Section 2.4 to make a brief discussion on the additional computational cost.

6. Following the comments of AnonReviewer2, we redrew most of the figures with larger fonts.

7. We made further tuning on baseline methods including Adam-L4 and Adam-HD for the experiments of ResNets, which perform poorly in the original submission. The new results have been updated in the revised version.

8. We added the experimental results of all candidate optimisers for both ResNet-18 and ResNet-34.

9. We revised the typos based on the comments of AnonReviewer2 and AnonReviewer3.

10. We further improved our written and grammar with the help of AJE edit service.

Thank you so much.

---

### Decision · Program_Chairs · 2021-01-07
**Final Decision**

**Decision:**

Reject

**Comment:**

The paper proposes an optimization framework that automatically adapts the learning rates at different levels of a neural network  based on hypergradient descent.  The AC and reviewers all found the approach interesting and promising and appreciate the author feedback.

We strongly encourage the authors to incorporate the points and additional results provided in their response to the reviewers.

Additionally, some concerns remain to be addressed regarding initialization of hyper-parameters of combination weights. Specifically it would be important to further investigate the impact of such initialization on the optimization performance. Furthermore, additional experiments with other network optimizers would be valuable as pointed out in the reviews.